# Characterisation of the long-term physical and mental health consequences of SARS-CoV-2 infection: A systematic review and meta-analysis protocol

Paul Kuodi 🔵 *, Yanay Gorelik◉, Michael Edelstein◉

Azrieli Faculty of Medicine, Bar-Ilan University, Safed, Israel

◉ These authors contributed equally to this work.
* kuodipa@biu.ac.il

## Abstract

### Background

As of July 2021, there has been more than 185 million documented cases of the novel coronavirus (SARS-CoV-2) infections and more than 4 million deaths globally. Despite more than 90% of documented cases being classified as "recovered" from SARS-CoV-2 infection, a proportion of patients reported a wide variety of persisting symptoms after the initial onset or acute phase of the infection, often referred to as "Long Covid". As data on the symptomatology of post-acute SARS-CoV-2 infection gradually becomes available, there is an urgent need to organise and synthesise the data in order to define what constitutes Long Covid and assist with its management in clinical and community settings.

### Methods

This protocol follows the Preferred Reporting Items for Systematic Review and Meta-analysis Protocols (PRISMA-P) guidelines. A comprehensive literature search strategy will be developed in accordance with the Cochrane highly sensitive search guidelines. The following electronic databases will be searched for studies to include in the systematic review and meta-analysis: MEDLINE (via PubMed), Scopus, Google Scholar, Web of Science (Web of Knowledge), Science direct, EMBASE, Mednar, Psych INFO, and EBSCOhost. Dual screening will be applied at every screening stage. Two reviewers will independently screen titles, abstracts and full text of potentially eligible studies following the predefined inclusion and exclusion criteria in order to select studies to include in the review. As heterogeneity is anticipated between the included studies, data will be pooled in a meta-analysis using a random effects model. A clustering analytic approach will be applied to identify symptoms groupings and assign the symptoms into clusters. R statistical software will be used for the meta-analysis. Highly heterogenous data will be synthesised narratively. The studies will be assessed, for quality using quality assessment tools appropriate for each study design. Two reviewers will independently undertake the quality of studies assessments.

**Funding:** The authors received no specific funding for this work.

**Competing interests:** The authors have declared that no competing interests exist.

**Abbreviations:** COVID-19, Coronavirus Disease 2019; DVT, Deep Vein Thrombosis; GRADE, Grades of Recommendation Assessment, Development and Evaluation; MeSH, Medical Subject Headings; PRISMA-P, Preferred Reporting Items for Systematic Review and Meta-analysis Protocols; PROSPERO, Prospective Register of Systematic Reviews; SARS-COV-2, Severe Acute Respiratory Syndrome Coronavirus 2; SOB, Shortness of Breath.

## Dissemination plans

Findings of the systematic review will be disseminated through a peer-reviewed publication and presentation of findings at conferences, workshops and government and private sector stakeholder engagement meetings.

## Clinical trial registration

**PROSPERO registration number**: CRD4202126589. https://www.crd.york.ac.uk/prospero/display_record.php?ID=CRD4202126589.

## Introduction

By July 2021, there were more than 185 million documented cases of the novel coronavirus (SARS-CoV-2) infections and more than 4 million deaths globally [1]. The clinical course of SARS-CoV-2 infection ranges from asymptomatic to a fatal disease [2]. Despite more than 90% of the documented cases being reported as "recovered" from SARS-CoV-2 infections [1], a proportion of patients continue to report persistent mild to severe symptoms after the initial onset or acute phase of the infection [2]. Owing to its novelty, many aspects of the SARS-CoV-2 infection are still not well understood. Specifically, the nature, extent and pathogenesis of its post-acute phase sequelae are yet to be well understood [3].

There is currently no full consensus among the scientific community about the pathway leading from SARS-CoV-2 infection to long term health consequences, a phenomenon commonly referred to as Long Covid or Chronic Covid syndrome [4]. As yet, there is no agreement on the definition of post-acute COVID-19 [2]. An early attempt defined post-acute COVID-19 as the clinical manifestation of symptoms that extend beyond 3 weeks following the initial symptoms of COVID-19 and Chronic Covid Syndrome (Long Covid) as clinical symptoms after initial onset of symptoms that extend beyond 12 weeks [5]. The current definition of Long Covid has fundamental limitations which curtails its application in clinical settings. Firstly, the definition of Long Covid is broad and not tagged to specific symptoms that can help care providers come to a diagnosis of the condition. A wide range of symptoms are reported in literature with little attempt to identify which symptoms genuinely constitute a post-acute consequence of SARS-CoV-2 infection [6]. Secondly, some studies reported Long Covid symptoms on patients who were never tested for SARS-CoV-2 infection in the first place [7]. Thirdly, asymptomatic patients (which includes up to 50% of COVID-19 cases) have largely been ignored in the reporting as research is currently focussed on hospitalised symptomatic cases.

The symptoms most commonly reported are fatigue, dyspnoea, joint and chest pain [8]. In addition, organ specific symptoms primarily of the brain, heart and the lungs have been reported [9,10]. Apart from the physical health consequences of SARS-CoV-2 infections, a host of psychosocial consequences of SARS-CoV-2 infection have also been reported [2]. The long list of symptomatology of Long Covid, the course of the disease including timing of symptoms onset, as currently documented is not conducive to building a consensus on a specific case definition or towards its clinical diagnosis and management.

The proposed systematic review and meta-analysis will attempt to measure qualitatively and quantitatively, using available data, the consequences of SARS-CoV-2 infection on the physical, and mental health domains.

As data on the symptomatology of post-acute SARS-CoV-2 infection gradually becomes available, there is an urgent need to organise and synthesise the data in order to define what constitutes and what does not constitute Long Covid and assist with its management in clinical and community settings. The review seeks to answer the research question: "*What are the long-term physical and mental health impacts of SARS-COV-2 infection on patients?*".

## Methods

### Study design

This protocol follows the Preferred Reporting Items for Systematic Review and Meta-analysis Protocols (PRISMA-P) guidelines [11]. The systematic review topic has been registered on the International Prospective Register of Systematic Reviews (PROSPERO) platform: Number: CRD4202126589 [12].

### Search strategy

A search strategy will be developed in accordance with the Cochrane highly sensitive search guidelines [13]. The search strategy was developed after preliminary review of existing literature on the topic and in consultation with clinicians who routinely manage Long Covid patients. The search strategy will be piloted on PubMed database before conducting the literature search in different databases.

The search strategy will be developed using search terms and free-text words. Boolean operators 'AND' and 'OR' as follows; ("long covid" OR "chronic covid" OR "post covid syndrome" OR covid OR covid-19 OR "long haul" OR "post-acute covid") AND (cough OR pain OR "shortness of breath" OR dyspn* OR palpitations OR heart failure OR "heart attack" OR "transient ischemia" OR "myocardial infarction" OR arrythmia OR stroke OR "cardiovascular accident" OR DVT OR "thromb*" OR PE OR "pulmonary embo*" OR fatigue OR weakness OR "joint pain" OR arthralgia OR "loss of smell" OR anosmia OR ageusia OR "loss of taste" OR seizure OR "visual problems" OR diplopia OR "peripheral neuropathy" OR "loss of sensation" OR paraesthesia OR tinnitus OR diarrhoea OR "loss of appetite" OR "muscle pain" OR myalgia OR constipation OR nausea OR vomiting OR abdominal discomfort OR abdominal cramp OR insomnia OR sleeplessness OR poor sleep, OR impaired sleep, OR lack of sleep OR Decreased cognitive function OR delirium, OR acute generalized cerebral impairment, OR acute organic brain syndrome OR impaired consciousness, OR Seizure OR convulsion OR epilepsy OR Impaired vision, OR blind* OR depression OR anxiety). The search terms will be modified to include controlled descriptors (such as MeSH terms) and their synonyms. See S1 Appendix for the provisional search strategy. The literature search will be restricted to include studies published between March 2020 and March 2022.

### Information sources

The following electronic databases will be searched for studies to include in the systematic review and meta-analysis: MEDLINE (via PubMed), Scopus, Web of Science (Web of Knowledge), Science direct, Google Scholar, EBSCOhost, EMBASE, PsycINFO, Cochrane Library, and Mednar for grey literature.

A hand-search of reference lists of included studies will be conducted. International Severe Acute Respiratory and Emerging Infection Consortium (ISARIC) and other consortiums with ongoing studies on Long Covid will be consulted to identify potential studies for inclusion in the review. Where necessary, expert opinion will be sought from clinicians who routinely manage Long Covid patients.

## Study selection

The review question will guide the inclusion and exclusion criteria for studies to include in the systematic review and meta-analysis. To be eligible for inclusion the systematic review and meta-analysis, studies will have to be original research articles of peer-reviewed observational studies; cohort studies, cross-sectional studies, and case-control studies. Other study designs including case reports, case series, RCTs and reviews of all kinds will be excluded. To be included in the study, original research articles will have to report on participants who tested for SARS-CoV-2 through serological tests or molecular tests irrespective of inclusion or exclusion of control groups. Furthermore, studies will be included only if they report self-reported or verified symptoms of SARS-COV-2 infection after symptoms of the initial SARS-COV-2 infection have subsided, or new symptoms not SARS-COV-2 infection related or ongoing SARS-COV-2 infection symptoms after the acute phase. No time restriction will be applied on the time of onset of the symptoms provided they are symptoms appearing on recovered patients. Studies reporting on the mental health consequences of SARS-COV-2 infection will also be included.

Finally, we will restrict the literature search to only studies published in English language with no restriction of geographical locations.

## Screening and data extraction

Dual screening will be applied at every screening stage. Two reviewers will independently screen titles, abstracts and full text of potentially eligible studies following the predefined inclusion and exclusion criteria in order to select studies to include in the review. The two reviewers will in addition independently extract data from included studies and grade the quality of evidence. At every stage of screening and quality of evidence grading, a third reviewer will be consulted if disagreements arise between the first and the second reviewers.

Data will be extracted using a data extraction tool designed specifically for this study and developed with considerations from the instructions provided by the Cochrane Collaboration [14]. In order to identify the data items to be extracted, the proposed tool considers four elements for data extraction: 1) details of the study (article title; journal title; authors; country of the study; language; publication year); 2) methodological characteristics (study design; study objective or research question or hypothesis; sample characteristics (e.g. sample size; sex; age, ethnicity; groups and controls; follow-up duration; validation of measures; statistical analyses); 3) main findings, and 4) conclusions.

## Study outcomes

Two categories of outcomes of SARS-CoV-2 infection will be considered in this systematic review and meta-analysis 1) Physical Health Outcomes of SARS-CoV-2 infections (Respiratory outcomes- Cough, SOB, dyspnoea), (Cardiovascular outcomes-palpitations, Heart attacks, arrythmias stroke, DVT, transient ischemia, pulmonary emboli, myocardial infarction, easy bleeding), (Musculoskeletal outcomes- fatigue, muscle weakness, joint pain, myalgia, arthralgia), (Neurological outcomes- syncope, Dizziness, seizures, tremors, diplopia, insomnia, ageusia, anosmia, tinnitus, loss of balance, pins and needles, loss of sensation, pain and discomfort, cognitive function deterioration), (Gastrointestinal outcomes- Abdominal pain, loss of appetite, constipation, diarrhoea, nausea and vomiting). 2) Mental Health Outcomes of SARS--CoV-2 infections- anxiety, depression, loss of selfcare ability.

## Study quality assessment

The studies will be assessed for quality using quality assessment tools appropriate for each study design. The Newcastle-Ottawa Scale (NOS) quality assessment tool will be used to assess quality of cohort studies, and The Joanna Briggs Institute (JBI) Critical Appraisal Checklist will be used to assess quality of analytical cross-sectional studies. Two reviewers will independently undertake the quality of studies assessments and risk of bias scoring. Any discrepancy in the quality assessed by the reviewers will be resolved through discussion before arriving at a consensus. Publication bias will be assessed using funnel plots if at least 10 studies are included in the meta-analysis [15] and the findings reported as appropriate.

## Data analysis

As it is anticipated that included studies will be heterogeneous given the diversity in the population of included participants, data will be pooled in a meta-analysis using a random effects model. Data measured with similar tools will be pooled together, where the measurement tools used varies widely, data will be synthesised narratively. R and STATA statistical software will be used for all data analysis. Highly heterogeneous data will be synthesised narratively. Heterogeneity will be assessed in the meta-analysis using Higgins & Thompson's $I^2$ Statistic and Predictive Intervals (PIs) with significance defined at the 10% α-level and quantified with the Higgins' $I^2$ statistic [15]. $I^2$ statistics of between 25% and less than 50% will be considered low heterogeneity, $I^2$ statistics of between 50% and less than 75% will be considered moderate heterogeneity and $I^2$ statistics of greater than 75% will be considered substantial heterogeneity.

Where data will be sufficient, subgroup analyses will be conducted by study design, participants characteristics, and by the clusters of outcomes of interest. If data will allow, unsupervised k-means clustering approach, will be applied to identify symptoms groupings and assign the symptoms into clusters. Sensitivity analyses will be carried out to examine the effect of risk of bias on the pooled estimates and the timing of Long Covid symptoms reporting to refine the definition. Studies with high risk of bias scores will be excluded from the meta-analysis. In the event that heterogeneity will be very high, the data will be synthesised narratively. Pre-existing health conditions will be put into consideration when computing frequency of reported symptoms.

The Grades of Recommendation Assessment, Development and Evaluation (GRADE) approach will be used to assess the certainty of the evidence [16]. Meta-analysis will be performed using R-statistical software version 4.1.0 and STATA V.16 software [17].

## Patient and public involvement

Patients will not be involved in the conduct of this systematic review and meta-analysis.

## Ethics and dissemination

The systematic review and meta-analysis will utilise publicly available literature and data. Hence, it will not require ethical approval.

## Discussion

In many countries, the race to contain the SARS-COV-2 pandemic through mass vaccination of the population is still underway. Emerging evidence shows that SARS-COV-2 infection has long term health consequences in many domains of health. Recovery from SARS-COV-2 infection takes a protracted path in a group of SARS-COV-2 infected patients, a condition that

is now widely recognised as Long Covid. Consequently, a new public health problem is now emerging in the wake of SARS-COV-2 pandemic.

The characterisation of Long Covid, its frequency, severity, signs and symptoms and health outcomes are still under investigation. Yet evidence-based data is urgently needed to inform the health planning for the new Long Covid public health problem. This systematic review attempts to collate available data on this important subject in order to inform global Long Covid health policy directions.

## Strengths and limitations of this study

- The systematic review and meta-analysis will be conducted and reported according to the guidelines of Preferred Reporting Items for Systematic Reviews and Meta-Analyses (PRISMA).

- A comprehensive literature search strategy has been developed. In addition, research organisations and consortiums with ongoing studies on long Covid will be consulted to identify potential studies for inclusion in the review.

- Literature search will be restricted to studies published in English language.

- The strength and quality of evidence from the review will be dependent on availability and quality of data on the review topic

## Supporting information

**S1 Appendix. Search strategy.**
(DOCX)

**S2 Appendix. PRISMA-P checklist.**
(DOCX)

## Author Contributions

**Conceptualization:** Paul Kuodi, Michael Edelstein.

**Data curation:** Paul Kuodi.

**Project administration:** Yanay Gorelik.

**Resources:** Paul Kuodi, Michael Edelstein.

**Supervision:** Michael Edelstein.

**Validation:** Paul Kuodi, Yanay Gorelik, Michael Edelstein.

**Visualization:** Paul Kuodi, Yanay Gorelik.

**Writing – original draft:** Paul Kuodi.

**Writing – review & editing:** Paul Kuodi, Yanay Gorelik, Michael Edelstein.

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
