## [Decision Letter · Decision Letter 0]

15 Feb 2022

PONE-D-21-25002Characterisation of the long-term physical and mental health consequences of SARS-CoV-2 infection: a systematic review and meta-analysis protocolPLOS ONE

Dear Dr. Paul Kuodi

Thank you for submitting your manuscript to PLOS ONE. After careful consideration, we feel that it has merit but does not fully meet PLOS ONE’s publication criteria as it currently stands. Therefore, we invite you to submit a revised version of the manuscript that addresses the points raised during the review process. There are some suggestion from the reviewers. Please follow the suggestion. 

We look forward to receiving your revised manuscript.

Kind regards,

Rizaldy Taslim Pinzon

Academic Editor

PLOS ONE

Journal Requirements:

2. Thank you for stating the following financial disclosure: "3000"

3. Thank you for stating the following in your Competing Interests section: "No competing interests" 

Reviewers' comments:

Reviewer's Responses to Questions

**Comments to the Author**

1. Does the manuscript provide a valid rationale for the proposed study, with clearly identified and justified research questions?

Reviewer #1: Yes

Reviewer #2: Yes

Reviewer #3: Yes

Reviewer #4: Partly

2. Is the protocol technically sound and planned in a manner that will lead to a meaningful outcome and allow testing the stated hypotheses?

Reviewer #1: Yes

Reviewer #2: Yes

Reviewer #3: Yes

Reviewer #4: Yes

3. Is the methodology feasible and described in sufficient detail to allow the work to be replicable?

Reviewer #1: Yes

Reviewer #2: Yes

Reviewer #3: Yes

Reviewer #4: Yes

4. Have the authors described where all data underlying the findings will be made available when the study is complete?

Reviewer #1: No

Reviewer #2: Yes

Reviewer #3: Yes

Reviewer #4: Yes

5. Is the manuscript presented in an intelligible fashion and written in standard English?

Reviewer #1: Yes

Reviewer #2: Yes

Reviewer #3: No

Reviewer #4: Yes

6. Review Comments to the Author

You may also provide optional suggestions and comments to authors that they might find helpful in planning their study.

Reviewer #1: The subject is good.

To the authors: Regarding the second outcome, Mental health outcome (Depression, anxiety) different studies may measure these outcomes differently or using different tools. How are you going to deal with this?

You may consider looking at the Grey literature as a means to reduce on the publication bias.

Its is good to specify the tools that you will use to assess the quality of the studies before hand (Of course different study designs require different tools)

Reviewer #2: Comments to the authors

This paper focuses on an important global problem. It aims to synthesise existing data to examine the long-term physical and mental health impact of SARS-COV-2 infection on patient. This paper would make important contributions to the ongoing global efforts to manage covid-19 infection. Some parts of the paper still require further work and specific comments are provided below:

1) Abstract

- The authors should also consider searching for relevant papers through EMBASE and PsycINFO databases.

- Also specify whether you intend to apply any geographical or language restrictions.

- What publication year/period do you intend to cover? e.g. 2020, 2021, etc? Mention it here.

2) Methods

- Line 96: What does the author mean by social health domain? This seems out of place since this paper is about physical and mental health impact of Covid-19.

- Lines 113 to 125, search strategy: here, the authors listed the search terms they intend to use. How did they identify the Covid symptoms listed here? Was it through published key literature and/or through discussions with librarians, experts, etc? This should be stated.

But what if the patients have symptoms other than those listed in the search terms? It would be a good idea for the authors to seek external input/feedback on this e.g. from clinicians.

- Line 125: Appendix 2? No other appendix was mentioned prior to this one. The authors should crosscheck the numbering – edit or mention appendix 1 in the appropriate place.

- Line 130 to 132, information sources: also consider searching EMBASE and PsycINFO, specify what publication years/periods you intend to cover and whether there will be geographical and language restrictions.

- Lines 145–150: in line with the study title, the authors should clearly indicate when they are referring to physical vs mental symptoms. For example, “…… self-reported or verified physical symptoms of SARS-COV-2 infection……..”.

- Line 164: host institution of the study may not be necessary. Consider extracting information about various definitions of post-acute covid, e.g. long covid or chronic covid.

- Line 181 to 182, study quality assessment: “The studies will be assessed for quality using quality assessment tools appropriate for each study design”. This statement is not sufficient as it does not tell readers which specific tool(s) you intend to use to undertake this task. Be specific.

- Line 187, descriptive analysis and meta-analysis: “data analysis” or “statistical analysis” would be a more appropriate name for this section. The authors should also consider conducting sensitivity analysis, if possible, to examine the effect of the definitions of post-acute covid e.g. >3 weeks (long covid) vs >12 weeks (chronic covid) vs both. This is important in view of lack of consensus regarding case definition which the authors mentioned in the introduction section.

Statistical analysis should also explore the presence/absence of pre-existing conditions. If there were pre-existing physical or mental health conditions, the persisting symptoms (during the post-acute phase) may not be exclusively due to the covid infection.

Reviewer #3: Authors to critically read through and make corrections to use of language. Authors could find useful other scientific research sites as contained in the attached document.

Reviewer #4: 1. In the space provided for the research question, please write it out in full.

2. Is the figure indicated (3000) expected funding? If so, please write the financial disclosure statement in full.

7. PLOS authors have the option to publish the peer review history of their article (what does this mean?). If published, this will include your full peer review and any attached files.

Reviewer #1: No

Reviewer #2: No

Reviewer #3: **Yes: **Mufutau Yunusa

Reviewer #4: No

---

## [Author Response · Author response to Decision Letter 0]

15 Mar 2022

Reviewer Reviewer Comments Responses to reviewer comments 

#1 Regarding the second outcome, Mental health outcome (Depression, anxiety) different studies may measure these outcomes differently or using different tools. How are you going to deal with this? 

We plan to pool outcomes from different studies only if they were measured with similar tools. Where different tools are used, we will synthesize the results narratively. 

(Line 204-205)

You may consider looking at the Grey literature as a means to reduce on the publication bias. 

We have added Mednar database to source Grey literature. (Lines 136-137)

Its is good to specify the tools that you will use to assess the quality of the studies before hand (Of course different study designs require different tools) 

The Newcastle-Ottawa Scale (NOS) quality assessment tool will be used to assess quality of cohort studies, and The Joanna Briggs Institute (JBI) Critical Appraisal Checklist will be used to assess quality of analytical cross-sectional studies.

(Line 194-196)

#2 

The authors should also consider searching for relevant papers through EMBASE and PsycINFO databases. 

EMBASE and Psych INFO have been added to the list of databases to be searched. (Line 140-141)

Also specify whether you intend to apply any geographical or language restrictions 

We will restrict the literature search to only studies published in English language with no restriction of geographical locations (Line 158-159)

What publication year/period do you intend to cover? e.g., 2020, 2021, etc? Mention it here 

The literature search will be restricted to include studies published between March 2020 and March 2022. 

(Line 131-133)

What does the author mean by social health domain? This seems out of place since this paper is about physical and mental health impact of Covid-19. 

The social health domain has been removed 

(Line 97)

search strategy: here, the authors listed the search terms they intend to use. How did they identify the Covid symptoms listed here? Was it through published key literature and/or through discussions with librarians, experts, etc? This should be stated. 

The search strategy was developed after preliminary review of existing literature on the topic and in consultation with clinicians who routinely manage Long Covid patients. (Line 112-113)

But what if the patients have symptoms other than those listed in the search terms? It would be a good idea for the authors to seek external input/feedback on this e.g. from clinicians. 

We are working in collaboration with clinicians who routinely manage Long Covid Patients 

Appendix 2? No other appendix was mentioned prior to this one. The authors should crosscheck the numbering – edit or mention appendix 1 in the appropriate place. 

Appendix 2 has been amended to appendix 1

(Line 131)

information sources: also consider searching EMBASE and PsycINFO, specify what publication years/periods you intend to cover and whether there will be geographical and language restrictions.

EMBASE and Psych INFO have been added to the list of databases to be searched

 in line with the study title, the authors should clearly indicate when they are referring to physical vs mental symptoms. For example, “…… self-reported or verified physical symptoms of SARS-COV-2 infection……..”. 

We are synthesizing data for self-reported symptoms and for verified symptoms. During analysis, we will separately synthesize self-reported and verified symptoms during. (Line 140-141) 

host institution of the study may not be necessary. Consider extracting information about various definitions of post-acute covid, e.g., long covid or chronic covid. 

Host institution of the study has been removed (Line 176). 

study quality assessment: “The studies will be assessed for quality using quality assessment tools appropriate for each study design”. This statement is not sufficient as it does not tell readers which specific tool(s) you intend to use to undertake this task. Be specific. Tools to be used for quality assessment have been listed accordingly. (Line 194-196)

descriptive analysis and meta-analysis: “data analysis” or “statistical analysis” would be a more appropriate name for this section. The authors should also consider conducting sensitivity analysis, if possible, to examine the effect of the definitions of post-acute covid e.g. >3 weeks (long covid) vs >12 weeks (chronic covid) vs both. This is important in view of lack of consensus regarding case definition which the authors mentioned in the introduction section.

“Descriptive analysis and meta-analysis” replaced with “data analysis” as suggested. 

(Line 201). 

Sensitivity analyses will be carried out to examine the effect of risk of bias on the pooled estimates and the timing of Long Covid symptoms reporting in order to refine the definition (Line 214-216) 

Statistical analysis should also explore the presence/absence of pre-existing conditions. If there were pre-existing physical or mental health conditions, the persisting symptoms (during the post-acute phase) may not be exclusively due to the covid infection Statistical analysis will be conducted to explore the presence/absence of pre-existing conditions in cases where data is reported. Studies that do not pre-existing conditions will be reported. (Line 219-219)

#3 

Authors to critically read through and make corrections to use of language. Authors could find useful other scientific research sites as contained in the attached document 

We have formatted the manuscript according to PLOS one Journal specifications

In the space provided for the research question, please write it out in full. 

Research question was written in full as suggested 

Is the figure indicated (3000) expected funding? If so, please write the financial disclosure statement in full. 

No specific funding was received for this study

---

## [Editor Report · Decision Letter 1]

17 Mar 2022

Characterisation of the long-term physical and mental health consequences of SARS-CoV-2 infection: a systematic review and meta-analysis protocol

PONE-D-21-25002R1

Dear Dr. Paul Kuodi 

We’re pleased to inform you that your manuscript has been judged scientifically suitable for publication and will be formally accepted for publication once it meets all outstanding technical requirements.

Kind regards,

Rizaldy Taslim Pinzon

Academic Editor

PLOS ONE
---

## [Editor Report · Acceptance letter]

25 Mar 2022

PONE-D-21-25002R1 

Characterisation of the long-term physical and mental health consequences of SARS-CoV-2 infection: a systematic review and meta-analysis protocol 

Dear Dr. Kuodi:

I'm pleased to inform you that your manuscript has been deemed suitable for publication in PLOS ONE. Congratulations! Your manuscript is now with our production department. 

Kind regards, 

on behalf of

Dr. Rizaldy Taslim Pinzon 

Academic Editor

PLOS ONE